# Individual and Contextual Risk and Protective Factors for Suicidal Thoughts and Behaviors among Black Adolescents with Arrest Histories

**DOI:** 10.3390/children9040522

**Published:** 2022-04-06

**Authors:** Camille R. Quinn, Erinn B. Duprey, Donte T. Boyd, Raven Lynch, Micah Mitchell, Andrew Ross, Elizabeth D. Handley, Catherine Cerulli

**Affiliations:** 1College of Social Work, The Ohio State University, Columbus, OH 43210, USA; boyd.465@osu.edu (D.T.B.); lynch.389@osu.edu (R.L.); mitchell.2074@osu.edu (M.M.); 2Mt. Hope Family Center, Department of Psychology, School of Arts and Sciences, University of Rochester, Rochester, NY 14627, USA; eduprey@ur.rochester.edu (E.B.D.); andrew.ross@rochester.edu (A.R.); elizabeth_handley@urmc.rochester.edu (E.D.H.); 3Department of Psychiatry, University of Rochester Medical Center & Susan B. Anthony Center, University of Rochester, Rochester, NY 14642, USA; catherine_cerulli@urmc.rochester.edu

**Keywords:** Black youth, suicide, positive parenting, arrests

## Abstract

Black adolescents in the United States have experienced an increase in suicidal thoughts and behaviors (STBs). Since Black adolescents are overrepresented in the youth punishment system, more research is needed to investigate correlates of STBs for this population. The purpose of this paper is to explore and establish correlates of individual, family, and community risk and protective factors and their relationship to lifetime STBs in a national sample of Black youth with arrest histories. Guided by an intersectional eco-behavioral lens, we investigated individual, family and contextual risk and protective factors for STBs among a national sample of justice-involved Black youth aged 12–17 with a history of arrest (*n* = 513). We used logistic regression models to test risk and protective factors for STBs. Among the sample, 9.78% endorsed suicidal ideation, and 7.17% endorsed a previous suicide attempt. Further, gender (female) and depression severity were risk factors for STBs, while positive parenting and religiosity were protective factors for STBs. School engagement was associated with lower levels of suicidal ideation. The findings suggest suicide prevention and intervention efforts should identify developmentally salient risk and protective factors to reduce mental health burden associated with STBs and concurrent alleged law-breaking activity of Black youth.

## 1. Introduction

Suicide continues to be a significant public health issue and is the second leading reason for death of adolescents and young adults in the United States [1]. A more recent trend includes the increasing rate of suicidal behaviors among Black American youth [2]. In addition, suicide has sweeping consequences, impacting parents, caregivers, family members and friends of those who attempt suicide or die by suicide [3]. The recent deaths of Black young adults, including public figures such as Chelsie Kryst, former Miss America winner, “Walking Dead” actor Moses Moseley, and Ian Alexander Jr. (son of Academy Award-winning actress Regina King) all died by suicide within weeks of each other [4]. This rise in suicide deaths among young Black people has been described as a first-time occurrence in history [4], and national trends indicate that although rates of suicidal ideations and plans are decreasing, rates of suicide attempts are increasing among Black adolescents [5,6,7]. Consequently, more research is needed to identify risk and protective factors and other correlates of STBs among Black youth and young adults [8]. Accordingly, the Congressional Black Caucus (CBC) established the Emergency Taskforce on Black Youth Suicide and Mental Health and called for research to identify risk and protective factors for STBs among Black youth in 2019 [9,10]. Moreover, the issue of suicide among Black youth, including males, overrepresented in the youth punishment system (stereotypical terms such as “juvenile offenders” and “juvenile justice system” promote stigma, and some use terms like “hypercriminalization” to describe the manner by which boys have been stigmatized and labeled as deviant and criminal, so they have been deliberately changed to “youth” and “youth punishment system” throughout this paper [11,12,13,14,15]), including youth with criminal activity histories, is of even greater concern.

There are racial and ethnic differences in the immediate risk factors for suicide. Lee and Wong [16] conducted a study with data from the National Death Reporting System (NVDRS), and their findings suggest that white youth were more likely to have a mental health diagnosis and treatment before suicide compared to other racial and ethnic groups. They noted Black youth were less likely than other racial and ethnic groups to have had a prior suicide attempt before their death, and to have had prior suicide ideation when they were compared to white youth. Further, Black youth were more likely to have a recent difficulty with law enforcement, which contributed to their death compared to Native American youth. The concern becomes greater when the overrepresentation of Black youth in the punishment system is considered even as arrest rates for young people are at their lowest in 40 years, especially for boys [17]. Despite the presence of national estimates of suicidal behavior (e.g., ideation, attempt and death by suicide) among youth in the punishment system, it is imperative to focus on the role of STBs among this population. The purpose of this paper is to explore and establish correlates of individual, family, and community constructs and their relationship to lifetime suicidal thoughts and behaviors in a national sample of Black youth with histories of involvement in the youth punishment system.

### 1.1. Theoretical Framework—Intersectionality and Bio-Ecological Model

The context of suicide and its impact on developmental outcomes in the lives of Black adolescents is becoming a common phenomenon. Intersectionality theory focuses on how the experiences of marginalized people exist in multiple forms of interlocking aspects of social oppressions and the toll they exert on people of color. The stress of trauma creates cumulative disadvantage linked to barriers such as racism, sexism, and other forms of oppression. Black adolescents, especially girls and young women, experience multiple oppressions that reinforce each other creating new categories of suffering [18,19,20]. Their experiences are instead embedded within these identities that exist within multiple environments with varying positions of influence. We use intersectionality as an organizing framework to highlight the intersecting identities and the cumulative effect that impact the health and mental health of Black adolescents [21,22]. As a result, it is important to take an intersectional approach regarding suicide and its impact on both Black girls and boys involved with the youth punishment system [23]. We also used a bio-ecological model to frame this study to investigate suicide among Black youth [24] characterized by distinct and intersecting risk and protective factors that contribute to their suicide risk [25]. Specifically, we were interested in risk and protective factors at the individual level (depression severity, substance use), the family level (the role of parenting), and the community level (the role of school engagement, activities). This model is needed to inform both intervention and prevention efforts to halt the spiking rates of Black youth suicide.

### 1.2. Depression, Substance Misuse and Suicidal Behavior

STBs are more common among youth in the punishment system versus those in the general population [26], which is linked to risk factors that are common among this population [26]. Scholars have noted that more than two-thirds of youth in detention facilities have one or more mental and/or substance use disorders [27]. Similarly, youth on probation (in the community with adjudicated cases by a judge) who reported STBs were more likely to do so if they also reported mental health and substance misuse issues [28]. In studies of pre-adjudicated youth in the community (pre-adjudication occurs before a judge reviews and settles a delinquency case), the prevalence of suicide attempts in the past month ranged between 1.4% to 2.9% [29,30,31], while lifetime attempts ranged from 9.9% to 13.2% [30,31,32]. Other studies have noted variation in factors associated with suicide for this population. Teplin et al. [27] noted that ACEs are also risk factors for suicidality among youth with a history of arrest [33,34,35,36]. One study with adolescents sent to the Florida Department of Juvenile Justice noted that reports of aggression and impulsivity explained the reason why multiple adverse childhood experiences (ACEs) that have been defined as the merging of “epidemiologic and neurobiological evidence of the effects of childhood trauma” (e.g., physical or sexual abuse, neglect, exposure to or witnessing violence, and parental or other family member who has been incarcerated) were associated with an increased risk for suicide attempts [37]. Further, depression and delinquency tend to be common and co-occurring symptoms among adolescents [38]. Although this co-occurrence predicts poorer mental health outcomes [39], it is uncertain if it also predicts worse delinquency outcomes. Moreover, empirical work highlights the severity of psychological distress of youth involved with the punishment system, but less is known about Black youth in the community and the influence of individual and contextual risk and protective factors and STBs.

### 1.3. Protective Factors for Suicidal Thoughts and Behavior

Scholars have identified religiosity as a protective factor for suicide, especially for adults and Black people [40,41]. Specifically, religiosity is a multifaceted concept like external activities (i.e., attending church services) and internal resources (i.e., spiritual or religious beliefs, coping and praying) [41,42]. High involvement in religious activities and spiritual well-being are both protective factors that may be particularly pertinent for Black youth and young adults, given that adolescence has been identified as a sensitive period for spiritual development [43,44,45,46,47]. Few studies explore how religion and spirituality relate to delinquency among Black youth, and even fewer studies explore the relationship to delinquency among these youth, and even fewer explore the correlation between religiosity and suicidality of justice-involved Black youth and young adults. Of the published studies, existing evidence suggests that religiosity and spirituality indeed play a protective role in the lives of Black youth against injurious behaviors, including delinquency and suicidality [48,49,50]. A meta-analysis of 62 studies by Kelly et al. [51] notes that religious involvement is inversely correlated with delinquent behaviors for both Black and white youth. Cole-Lewis et al. [52] echoed these findings, indicating that lower levels of suicidality were associated with organizational religiosity among Black and white youth experiencing interpersonal problems. Although religion and spirituality have been documented as protective factors against suicidality and delinquency for Black youth, it is worth noting that the rigid ideals held within faith communities have hindered some youth from seeking mental health services [53,54,55,56]. A focus group study of Black teens experiencing depression noted a lack of information about mental illness shared within religious institutions, as well as a heavy dependence upon prayer, a major cultural barrier to treatment engagement [55].

Many youth enjoy both religious and school activities based on the benefits from their participation, including those in their community. Specifically, youth who were able to take advantage of having an outlet such as community centers and activities such as sports or mentorship enjoy significant benefits [57]. For some youth, many problem behaviors could be associated with social challenges if youth mimic adverse behaviors they may witness at school or in the community if their delinquency is not curtailed [57,58], and this may also include suicidal behavior. In addition, when Black youth are removed from school and activities, they may view this as threatening to their identity, which could elicit adverse responses—such as victimizing behaviors as well as other belligerent acts—to the threat [59,60,61], while also increasing their contact with the youth punishment system.

Black families generally encourage stronger parent–child attachment [62,63], and parental support is even more important when youth are troubled or experiencing challenges and struggles [62,64]. Parents and caregivers’ support provides a solid influence in their children’s lives [62,65]. Further, the health and mental health, and wellness of children and youth’s parents and caregivers is an important factor in their overall functioning [62,66]. Specifically, when youth reported parental support such that they viewed the relationships with their parents as affirmative, Black youth endorsed variations in reporting suicidal ideations but not attempts [67]. Other studies of parent actions were positively associated with the protective effects for STBs among youth of color, including Black youth [68]. For example, in studies about parents and their child/ren’s education, the results also suggest a protective role against STBs [69,70,71]. Consequently, there should be comprehensive efforts to include the active involvement of parents, caregivers and family members to bolster positive parent–child relationships as a buffer against the risk factors associated with STBs for youth in the punishment system.

## 2. Current Study Aims and Hypotheses

Intersectionality theory and the bio-ecological model of human development, along with prior research on Black youth and young adults with histories and STBs guide the study aims and corresponding hypotheses. We aimed to identify risk and protective factors spanning from individual (e.g., mental health diagnoses) to contextual (school-based or family-based) factors (See Figure 1). Thus, we examined individual risk factors including sex, socioeconomic status, depression and substance misuse, individual protective factors (i.e., religiosity), and family (i.e., positive parenting), school (i.e., school engagement), and community levels (youth involvement in activities). We examined all risk and protective factors for three related outcomes: lifetime suicidal thoughts, lifetime suicidal plans, and lifetime suicide attempts. It was important to differentiate between these three outcomes, due to the consensus in the suicidology literature that the etiology and associated risk and protective factors may be different for STBs.

Among the individual factors, we hypothesized that girls would have higher rates of STBs, and that depression severity and substance misuse would also be associated with higher rates of STBs. Among the contextual factors, we hypothesized that school engagement, religiosity, participation in extracurricular activities, and positive parenting would all lessen the odds of STBs.

## 3. Materials and Methods

### 3.1. Participants

Participants were Black youth aged 12–17 who participated in the National Survey on Drug Use and Health (NSDUH) study from years 2014 to 2019 and who had prior history of arrest (*n* = 513). Of these youth, there were 31.38% (*n* = 161) who identified as female. The majority of participants (80.5%) had a family income less than $49,999 per year, and 59.5% of participants’ families participated in one or more government assistance program. In terms of poverty level, 51.1% of youth were from families who were below the federal poverty level, 26.5% of youth were from families with income up to two times the federal poverty threshold, and 22.4% of youth were from families who exceeded at least two times the federal poverty threshold.

### 3.2. Procedures

The present study is a secondary data analysis using NSDUH data from years 2014–2019 (for full NSDUH study procedures, see [72]). NSDUH is sponsored by the Substance Abuse and Mental Health Services Administration (SAMHSA) and is conducted each year to collect nationally representative data on drug use, mental health, and health behaviors in the general (i.e., non-institutionalized) population aged 12 and older. Individuals are selected for inclusion in the NSDUH based on a multistage stratified sampling design in all 50 states. Interviews are conducted by trained research staff using a handheld computer to record interview results. Parental consent and youth assent was collected before researchers interviewed individuals aged 12 to 17. An ACASI method was used to administer sensitive questions (i.e., about drug use).

### 3.3. Measures

**Lifetime Suicidal Thoughts and Behaviors.** Interviewers administered a youth depression module, via ACASI software (ACASI (audio computer-assisted self-administered interview) equipment technique allows standardization of the way in which questions are asked and who is asking them, and it eliminates interviewer interpretation of responses. The perceived anonymity of this type of interview may make respondents feel more at ease in reporting behaviors that are socially undesirable and less likely to embellish responses for socially desirable behaviors [73]), to all individuals between the ages of 12 to 17. Participants first responded to questions about depressive symptoms, including if they ever experienced a period lasting longer than a few days when most of the day they felt “sad, empty or depressed”, or were “very discouraged about how things were going in [their] life”, or had “lost interest in most things [they] usually enjoy like work hobbies, and personal relationships”. Following, youth were asked: “Did you ever think about committing suicide?”; “Did you make a suicide plan?”; and “Did you make a suicide attempt?” Suicidal ideation, planning, and attempts were coded as dichotomous variables with “0” indicating no presence of the symptom or behavior and “1” indicating presence of the symptom or behavior.

**Depression Severity.** Within the aforementioned depression module, youth were also asked to assess how much their depressive symptoms in the last 12 months interfered with four life domains: chores at home, school or work, family relationships, and their social life. For instance, one item was “The symptoms have disrupted your school work”, in which youth were instructed to rate this statement from “0” (not at all) to “10” (extremely). The NSDUH study team then recoded this variable from “1” (none; original category 0) to “5” (very severe; original category 10). This item was derived from the Sheehan Disability Scale (SDS; [74]). The maximum level of severity of impairment in any domain was used to assess depression severity in the present study.

**Substance Use.** Past year substance misuse was measured with an index that summed the presence of alcohol use, marijuana use, and cigarette use in the past month. Participants were asked the frequency that they used alcohol, marijuana, and smoked cigarettes in the past month. Responses were recoded so any frequency was given a score of “1” and no use was given a score of “0”. A sum score was then calculated that ranged from zero to three.

**Religiosity.** Youth religiosity was measured with a mean score of three items about religious participation and beliefs (α = 0.77). Items were “My religious beliefs are very important to me”, “My religious beliefs influence my decisions”, and “It is important that my friends share my religious beliefs”, with response options ranging from “1” (strongly disagree) to “4” (strongly agree).

**Positive Parenting.** Positive parenting was measured with a mean score of two items about parents’ supportive verbal behaviors (α = 0.85). Items included “During the past 12 months, how often did your parents let you know when you’d done a good job?” and “During the past 12 months, how often did your parents tell you they were proud of you for something you had done?” with response options ranging from “1” (always) to “4” (never). Both items were reverse scored so that higher scores represented more positive parenting, and then an average was calculated.

**School Engagement.** We assessed school engagement using a mean score on four items (α = 0.78). Items assessed how youth felt overall about going to school (1 = “you liked going to school a lot” to 4 = “you hated going to school”), how often they felt their schoolwork was meaningful (1 = “always” to 4 = “never”), how important they thought the things they learned in school were (1 = “very important” to 4 = “very unimportant”), and how interested they thought their classes were (1 = “very interesting” to 4 = “very boring”). All items assessed youths’ feelings in the past 12 months. Items were reverse scored before averaging so that higher scores reflected higher levels of positive school engagement.

**Activities.** An index of extracurricular activities was created that reflected youths’ participation in school-based, community-based, faith-based, and other activities. Adolescents were asked four questions about the frequency of their involvement in school-based activities (i.e., “During the past 12 months, in how many different kinds of school-based activities, such as team sports, cheerleading, choir, band, student government, or clubs, have you participated?”), community-based activities (“During the past 12 months, in how many different kinds of community-based activities, such as volunteer activities, sports, clubs, or groups have you participated?”), faith-based activities (“During the past 12 months, in how many different kinds of church or faith-based activities, such as clubs, youth groups, Saturday or Sunday school, prayer groups, youth trips, service or volunteer activities have you participated?”) and other activities (“During the past 12 months, in how many different kinds of other activities, such as dance lessons, piano lessons, karate lessons, or horseback riding lessons, have you participated?”). Response options ranged from “0” (none) to “3” (three or more). A sum score was calculated from the four items.

**Covariates.** Covariates included sex, coded as 1 = male and 2 = female, and SES risk, which was an index comprised of income (given a score of “1” if family income was below $20,000), poverty (given a score of “1” if participants’ family fell at or below the federal poverty threshold), and receipt of government assistance (given a score of “1” if participants’ family received assistance from government programs such as food stamps or cash assistance. The three items were summed so that a higher score represented greater socioeconomic risk.

### 3.4. Data Analysis

All analyses accounted for the complex survey structure of the NSDUH by using the Complex Samples utility in SPSS version 26, which allowed us to perform all analyses with the appropriate design/nesting variables and weights. First, univariate analysis wasconducted to investigate the associations connecting each of our hypothesized risk and protective factors, separately, with suicidal thoughts and behaviors. Following this, we tested three adjusted logistic regression models (i.e., in a multivariate analysis) that were run separately for each outcome (suicidal ideation, suicide planning, and suicide attempts). The adjusted logistic regression allowed us to determine the influence of each risk and protective factor while adjusting (i.e., controlling) for the other predictor variables in our model.

Missing data ranged from 0 to 19.1% depending on the study variable. Unadjusted and adjusted logistic regressions were modeled using the sample with complete data (i.e., listwise deletion).

## 4. Results

### 4.1. Preliminary Analyses

Descriptive statistics were first examined for all study variables (See Table 1). Among Black youth with a history of arrest, the weighted frequency for suicidal ideation was 9.78%, for suicide planning it was 6.48%, and for suicide attempts it was 7.17%.

### 4.2. Univariate Logistic Regression Models

See Table 2 for full results. Youth sex was associated with suicidal thoughts and behaviors such that boys were significantly less likely to exhibit ideation (OR = 0.10, *p* < 0.001), planning (OR = 0.12, *p* < 0.001) and attempts (OR = 0.10, *p* < 0.001) compared to females.

Depression severity was also associated with a significant increase in odds for suicidal ideation (OR = 2.57, *p* < 0.001), suicidal planning (OR = 2.36, *p* < 0.001), and suicide attempts (OR = 2.38, *p* < 0.001), as expected.

In terms of protective factors, higher levels of positive parenting were associated with lower levels of suicidal ideation (OR = 0.61, *p* < 0.01), planning (OR = 0.54, *p* < 0.001), and attempts (OR = 0.51, *p* < 0.001). Additionally, higher levels of school engagement were associated with lower levels of suicidal ideation (OR = 0.47, *p* < 0.01), planning (OR = 0.61, *p* < 0.05), and attempts (OR = 0.42, *p* < 0.01).

### 4.3. Multivariate Logistic Regression Models

See Table 3 for full results. The multivariate model included all independent variables entered simultaneously to test the associations with suicidal ideation, sex and depression severity remained the only significant factors (respectively: OR = 0.17, *p* < 0.01; OR = 2.51, *p* < 0.001). Sex, depression severity, and positive parenting were all significant predictors of suicide planning (respectively: OR = 0.30, *p* < 0.05; OR = 2.33, *p* < 0.001; OR = 0.52, *p* < 0.05). Finally, sex and depression severity were significantly associated with suicide attempts (respectively: OR = 0.85, *p* < 0.01; OR = 2.17, *p* < 0.001).

## 5. Discussion

The connection between suicide and involvement with the youth punishment system has been established, especially for adolescents who are detained or incarcerated [27,60,75]. Youth and criminal punishment system interventions often focus on the individual versus contextual and/or macro level factors associated with their behavioral outcomes. Further, there tends to be a primary focus on risk versus protective factors. Although risk factors are amendable, there remains the opportunity to consider protective factors (i.e., strengths and assets) that could be leveraged or promoted. The current study was guided by intersectionality theory [18,19,20] and the bio-ecological model to investigate lifetime STBs among Black adolescents with histories of arrest [24,25,32]. Results showed that both risk and protective factors across their bio-ecological context matter in the etiology of STBs. Specifically, the multivariate logistic regression indicated significant associations between sex, depression severity, and positive parenting with youths’ likelihood of STBs. Sex and depression severity were both significantly associated with suicidal ideation, planning and attempts, as expected. Further, even in the context of all other risk and protective factors, positive parenting emerged as a protective factor that decreased the odds of reporting suicidal planning for the adolescents in this study, which is consistent with other study findings [76,77,78]. This finding is important, as it highlights the significant role that families can play with regard to suicide prevention for this population.

Our study sample of general population youth ages 12–17 years included 9.78% who reported suicidal ideation, 6.48% who reported suicidal planning, and 7.17% who reported a suicide attempt over their lifetime. When we compare them to other populations of adolescents, the findings are mixed. For example, the National Comorbidity Survey noted suicidal behavior over the lifetime of youth ages 13–18 reported ideation (12.1%) and attempts (4.1%), respectively [26]. The rates are higher when the timeframe is restricted to the past year. The Youth Risk Behavior Survey (YRBS) included youth ages 15–19 years who reported higher rates of ideations (15.8%) and attempts (7.8%) in the past year [26,79]. In another study using YRBS data, 11.1% of youth reported suicidal planning in the past year among families living in a Mid-Atlantic public housing development [28,79]. However, for adolescents ages 12–18 years on probation in a Midwestern jurisdiction, 5.79% reported STBs (suicidal thoughts or behaviors, including suicidal ideation: attempts or thoughts to harm self) at the point they were assessed (in 2014, the agency that oversees the Courts utilized the Youth Assessment Screening Instrument (YASI) as the primary risk assessment and implemented it statewide; it comprised risk and protective indicators in 10 domains (Legal History, Family, School, Community and Peers, Alcohol and Drugs, Mental Health, Aggression, Attitudes, Skills, and Employment and Free Time) with 72 questions) [76]. A higher prevalence of suicidal ideation and planning is likely in the present study, since the Mid-Atlantic study only asked about suicidal planning in the last 12 months, whereas in the present study, it asked about suicidal behavior at some point in their lifetime. Further, the study in the Midwestern jurisdiction only asked about suicidal ideations and attempts versus planning. Additionally, of note, the national studies reported higher rates of ideation also comprised older populations than those in our study sample suggesting that STBs could be an issue that exacerbates over time. The findings in this study on the prevalence of ideation, planning as well as attempts, provides more detailed information about the STBs for Black youth in the general population with arrest histories. One could argue that Black adolescents in this study who did not report positive parenting (parental support) may experience more difficulties based on the convergence of their multiple and marginalized identities, including their involvement with and overrepresentation in the youth punishment system [62,75,80,81], especially since they reported such high rates of suicide attempts.

At the individual level of the bio-ecological model, we investigated sex, depression severity, and substance use as risk factors for youth’s lifetime STBs. Study findings were consistent with existing research about suicide risk [28,82,83]. Specifically, youth sex was significantly associated with STBs as boys were less likely to exhibit them than girls, which is consistent with national statistics regarding gender differences [84].

At the family level, we also found participants who reported protective factors, including higher levels of positive parenting, and were less likely to report suicidal ideations and attempts. This is consistent with other empirical work that underscore the power of parents’ roles in the lives of their children based on their levels of support and the quality of the relationships [63,64,65,67,85]. Previous studies with Black adolescents suggest noted variation in reporting ideations but not attempts, especially if they reported parental support, i.e., when they viewed that their parent relationships were positive [67]. From an intersectionality lens, it could be that Black adolescents may not want to be perceived in a manner that may lead them to be further marginalized. Specifically, they may feel comfortable indicating that they have thought about suicide, but some may not want their parents to know that they have actually made an attempt. One way to think of this is that Black adolescents may present both a public face (to their parents and other family members) and a private face (to those who may share their sentiments and feelings), to reduce the impact of a further marginalized identity.

We noted strong positive associations at the community level, where Black adolescents who reported higher levels of school engagement also reported lower levels of suicidal ideation. This is significant because many individuals involved in the youth punishment system often face the stigma of arrest. Moreover, if the arrest occurs in school and the result is detainment or incarceration, they face grave scrutiny and they experience stereotypes and stigmatization when they return to school [11,12,15,80]. This reflects the multiple oppressions associated with what intersectionality defines as the stress of trauma that operates like a triple jeopardy of barriers (racism, sexism) for Black adolescents making them more vulnerable to STBs. For example, youth in the punishment system are more likely to have increased educational, health (physical, mental, sexual), social, legal, and economic challenges than their non-system involved counterparts [86]. In addition, youths’ mental health problems are positively linked to the depth of their involvement with the punishment system for Black youth [87]. Overall, youth engaged in the youth punishment system demonstrate that experiences of adversity are related to poorer functioning over time [88] and greater mental health and substance-related needs [89].

### 5.1. Limitations

Overall, these results contribute to the knowledge about an understudied subpopulation, namely, Black youth with a history of arrest. Our work informs efforts to determine the best ways to modify the individual, family and contextual factors to prevent STBs in this population. Despite this, there are limitations to this study. The first is that it is a secondary dataset, limiting the ability to answer research questions beyond those posed by the original researchers. Related to this limitation, other relevant contextual risk factors could not be included due to lack of data. In particular, there is a potential for ACEs and racism, racial trauma and/or cultural resilience to be associated with STBs in this population. Similarly, the dataset does not provide information on the timing of STBs, only if they have ever occurred. Another limitation includes the single item used for youth depression severity in this study. Ideally, a more precise depression measure, including a clinical cut-off would have been more useful to identify symptomology among Black youth. As such, findings should be interpreted with caution. Finally, we explored direct relationships, but not interrelated independent variables or mediating pathways.

Future research should consider including further data collection focused specifically on a wide range of culturally relevant risk and protective factors for STBs, more mixed methods studies to provide contextual information about the risk and protective factor assessments, and listening to youth’s voices in the creation, implementation and testing of targeted interventions. Conducting mixed methods studies that incorporate both parents and caregivers’ views on positive parenting would be useful to develop training programs to enhance their skills in this area. In addition, future studies on suicide prevention and interventions with this population need to be prioritized by classifying specific risk and protective factors as well as age-related mechanisms related to Black youth suicidal behavior [90]. To implement effective suicide prevention programming, understanding targets for intervention is necessary [90], and such programming would benefit from incorporating parents to investigate treatment modalities specific to youth involved with the punishment system, e.g., healing-centered engagement, mindfulness, and multisystemic therapy. There is a need for further research using nonrandomized as well as randomized samples (with control groups) to more confidently establish the efficacy of these interventions. Moreover, developing training on positive parenting skills, to strengthen the bond between adolescents and their parents and caregivers is needed to lessen the odds of STBs.

### 5.2. Practice and Policy Implications

This study provides important current information for policymakers and practitioners. For policymakers, the study highlights the significance of school engagement as a protective factor for STBs, an activity often impacted by policies affecting school staffing and funding. Additionally, depression was a significant predictor of STBs for Black youth, and it was previously stated that white youth are more likely to have been treated for depression prior to a suicide attempt than youth of color, and specifically Black youth are less likely than their peers of other races and ethnicities to have expressed suicidal ideation or had a prior suicide attempt before dying by suicide [16]. As such, it is prudent to direct policy to increase and improve mental health services that reach Black youth and identify depression (including culturally-tailored measures) and other suicidal risk factors before any attempts, especially considering that for Black youth, their first attempt is often lethal.

For practitioners, the study highlights the importance of engaging parents and the parent–child relationship in treatment of depression and reported STBs, as sex, depression severity, and positive parenting were the only significant protective factors for suicide attempts remaining after the multivariate logistic regression. Additionally, practitioners should shift focus to reaching Black youth and identifying depression warning signs early so that Black youth, like white youth, can receive mental health treatment before expressing STBs, thus potentially preventing them all together.

## Figures and Tables

**Figure 1 children-09-00522-f001:**
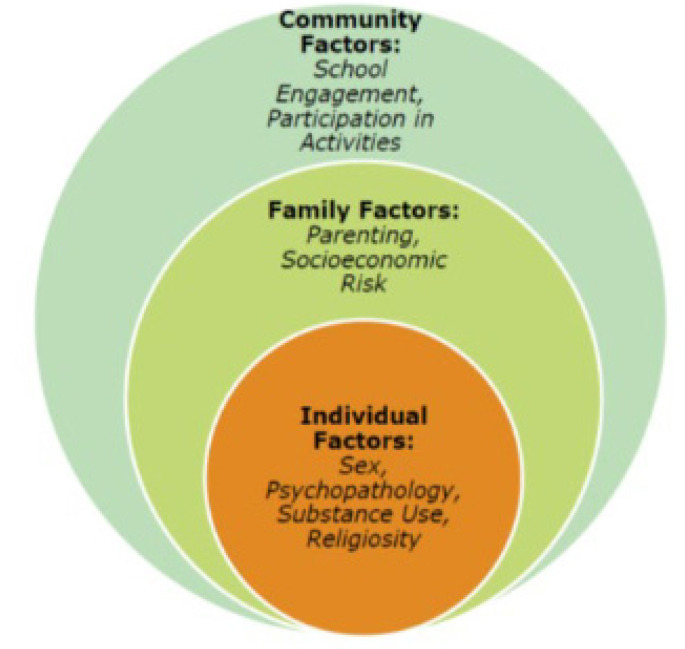
An Intersectional Bio-ecological Model of Black Youth Suicidal Thoughts and Behaviors (STBs).

**Table 1 children-09-00522-t001:** Descriptive statistics for study variables.

Predictor	Mean	Standard Error	Range	Unweighted *N*
Socioeconomic risk	1.50	0.07	0–3	415
Depression severity	0.48	0.07	0–5	513
Substance use	0.48	0.04	0–3	513
Positive parenting	3.25	0.05	1–4	505
School engagement	3.00	0.04	1–4	465
Activities	4.47	0.19	0–12	503
Religiosity	2.67	0.05	1–4	498

**Table 2 children-09-00522-t002:** Unadjusted logistic regression for STBs.

		Suicidal Ideation					Suicide Plan					Suicide Attempt			
Predictor	*N*	*B*	*SE B*	OR	95% CI OR	*N*	*B*	*SE B*	OR	95% CI OR	*N*	*B*	*SE B*	OR	95% CI OR
*Demographics*															
Sex (male)	495	−2.35	0.43	0.10 ***	[0.10, 0.04]	495	−2.14	0.47	0.12 ***	[0.05, 0.30]	494	−2.32	0.49	0.10 ***	[0.04, 0.26]
SES Risk	495	0.49	0.38	1.63	[0.75, 2.54]	495	0.05	0.48	1.06	[0.40, 2.76]	494	0.37	0.49	1.45	[0.55, 3.84]
*Individual risk factors*															
Depression severity	495	0.94	0.12	2.57 ***	[2.04, 3.24]	495	0.86	0.12	2.36 ***	[1.85, 3.02]	494	0.87	0.12	2.38 ***	[1.87, 3.03]
Substance use	495	−0.13	0.17	0.88	[0.63, 0.12]	495	−0.04	0.22	0.96	[0.62, 1.48]	494	−0.10	0.23	0.91	[0.58, 1.43]
*Protective factors*															
Positive parenting	493	−0.50	0.16	0.61 **	[0.44, 0.84]	493	−0.62	0.16	0.54 ***	[0.39, 0.74]	492	−0.68	0.18	0.51 ***	[0.36, 0.72]
School engagement	454	−0.76	0.23	0.47 **	[0.30, 0.74]	454	−0.49	0.20	0.61 *	[0.41, 0.92]	453	−0.86	0.27	0.42 **	[0.24, 0.73]
Activities	494	0.01	0.06	1.01	[0.89, 1.14]	494	0.04	0.07	1.04	[0.91, 1.18]	493	0.03	0.08	1.03	[0.88, 1.20]
Religiosity	488	0.00	0.19	1.00	[0.68, 1.46]	488	−0.06	0.21	0.94	[0.61, 1.44]	487	−0.01	0.23	0.99	[0.62, 1.57]

*Note:* OR = Odds Ratio (exponentiated *B*). Interactions were tested in separate models. * *p* < 0.05. ** *p* < 0.01. *** *p* < 0.001.

**Table 3 children-09-00522-t003:** Adjusted logistic regression for STBs.

	Suicidal Ideation (*n* = 449)				Suicide Plan (*n* = 449)				Suicide Attempt (*n* = 448)			
Predictor	*B*	*SE B*	OR	95% CI OR	*B*	*SE B*	OR	95% CI OR	*B*	*SE B*	OR	95% CI OR
*Demographics*												
Sex (male)	−1.75	0.53	0.17 **	[0.06, 0.50]	−1.19	0.51	0.30 *	[0.11, 0.84]	−1.38	0.47	0.25 **	[0.10, 0.65]
SES Risk	0.39	0.66	1.48	[0.40, 5.52]	−0.54	0.62	0.59	[0.17, 2.05]	0.07	0.75	1.08	[0.24, 4.88]
*Individual risk factors*												
Depression severity	0.92	0.16	2.51 ***	[1.82, 3.45]	0.85	0.17	2.33 ***	[1.65, 3.30]	0.77	0.16	2.17 ***	[1.58, 2.97]
Substance use	−0.56	0.44	0.57	[0.24, 1.38]	−0.22	0.32	0.80	[0.42, 1.54]	−0.47	0.36	0.63	[0.30, 1.30]
*Protective factors*												
Positive parenting	−0.34	0.31	0.71	[0.38, 1.33]	−0.66	0.29	0.52 *	[0.29, 0.93]	−0.55	0.33	0.58	[0.30, 1.13]
School engagement	−0.34	0.32	0.71	[0.38, 1.35]	0.49	0.46	1.63	[0.65, 4.13]	−0.26	0.34	0.77	[0.39, 1.52]
Activities	0.13	0.08	1.13	[0.97, 1.32]	0.14	0.09	1.15	[0.96, 1.39]	0.14	0.09	1.15	[0.95, 1.37]
Religiosity	0.52	0.28	1.68	[0.97, 2.92]	0.40	0.33	1.50	[0.78, 2.88]	0.37	0.42	1.45	[0.62, 3.36]

*Note:* OR = Odds Ratio (exponentiated *B*). * *p* < 0.05. ** *p* < 0.01. *** *p* < 0.001.

## Data Availability

The present study is a secondary data analysis using NSDUH data from years 2014–2019 (for full NSDUH study procedures, see [USDHHS, 2019]). NSDUH is a sponsored by the Substance Abuse and Mental Health Services Administration (SAMHSA) and is conducted each year to collect nationally representative data on drug use, mental health, and health behaviors in the general (i.e., non-institutionalized) population aged 12 and older.

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
