# Peer review of "Individual and Contextual Risk and Protective Factors for Suicidal Thoughts and Behaviors among Black Adolescents with Arrest Histories"

_children, 2022, doi:10.3390/children9040522_

Round 1

Reviewer 1 Report

The current study examined the link between risk and protective factors and suicidal thoughts and behaviors (STBs) among a sample of Black justice-involved youth (N = 513). Ultimately, the authors found that female gender and depressive symptoms were risk factors for SITBs. Furthermore, the authors also found that positive parenting and religiosity were protective factors.

Overall, the manuscript is well-written and makes a contribution to the extant literature. However, several issues warrant comment:

Introduction

  • The introduction is well-written and clear, laying out a solid foundation for the present study.
  • The section entitled “Depression, Substance Misuse & Suicidal Behavior” would benefit from fleshing out to discuss more about the results from studies for justice-involved youth and suicidal behavior.
  • Although the use of the term “punishment system” may be appropriate, many readers may be unfamiliar with the term. Orienting readers to the meaning of the term either through a footnote or a parenthetical would strengthen the manuscript.
  • The paragraph on parenting as a protective factor is underdeveloped, especially given that the authors’ have a major finding related to parenting. For example, the authors could incorporate more information regarding specific parenting practices as a protective factor.
  • The paragraph on intersectionality and the bioecological model could come prior to the discussion of risk factors, especially given that it would help introduce the notion that there are risk and protective factors at multiple levels of youths’ social ecologies.

Method

  • It would be helpful if the authors’ defined the acronym ACASI and the methodology or refer the reader to prior publications using NSDUH data/methods so that the methods of data collection are clear.
  • It would also be useful for the authors’ to clarify whether the NSDUH measures (e.g., religiosity, positive parenting, school engagement, and activities) were based on pre-existing measures of these constructs or created questions as well as which specific items from the NSDUH were used for the purposes of these analyses. This will help other researchers with replicability.

Results

  • The authors’ should add in information regarding missingness and what was done with missing data (e.g., listwise deletion, multiple imputation).
  • More rationale is needed regarding to the reasoning for running adjusted and unadjusted logistic regression models. It would also be helpful for the authors’ to add a sentence or two to orient the reader to the interpretation of odds ratios.
  • There appears to be a paragraph included in error beginning on line 286 with “This section may be divided by subheadings.”

Discussion

  • The authors should be cautious in overfocusing their discussion on factors that did not emerge as significant in their final multivariate logistic regression model. Variables that were statistically significant in the single logistic regression models could have emerged as significant due to spurious associations.
  • The authors’ description of their multivariate model should come at the beginning of the discussion and not towards the end.
  • Given that this present sample involves justice-involved youth, some attention to of the role that involvement plays in the present results would strengthen the Discussion section.
  • Given that a single-item measure was used for youth depression severity, this should be noted as a limitation.
  • It would also be great for the authors’ to have a separate future directions paragraph. For example, the authors’ suggest training on positive parenting skills as a future direction. How could this be fit into research on treatment modalities specific to justice-involved youth (e.g., multisystemic therapy)?

Author Response

REVIEWER 1

The introduction is well-written and clear, laying out a solid foundation for the present study.

The section entitled “Depression, Substance Misuse & Suicidal Behavior” would benefit from fleshing out to discuss more about the results from studies for justice-involved youth and suicidal behavior.

Fleshed out, revised and added content from studies with youth in the punishment system and suicidal behavior.

Although the use of the term “punishment system” may be appropriate, many readers may be unfamiliar with the term. Orienting readers to the meaning of the term either through a footnote or a parenthetical would strengthen the manuscript.

Inserted a footnote to defined “youth punishment system” with this statement:

Stereotypical terms like “juvenile offenders” and “juvenile justice system” promote stigma, so they have been deliberately changed to “youth” and “youth punishment system” throughout this paper [68-70]. And these citations/references:

Cole, H., & Cohen, R. (2013). Breaking down barriers: A case study of juvenile justice personnel perspectives on school reentry. Journal of Correctional Education, 64(1), 13–35.

Kubek, J. B., Tindall-Biggins, C., Reed, K., Carr, L. E., & Fenning, P. A. (2020). A systematic literature review of school reentry practices among youth impacted by juvenile justice. Children and Youth Services Review, 110, 104773.

Quinn, C. R., Boyd, D., Beaujolais, B., Hughley, A., Mitchell, M., & DiClemente, R. J., & Voisin, D. R. (2022). Perceptions of sexual risk and HIV/STI prevention among Black adolescent girls in a detention center: An investigation of the role of parents and peers. Journal of Racial and Ethnic Health Disparities.

The paragraph on parenting as a protective factor is underdeveloped, especially given that the authors’ have a major finding related to parenting. For example, the authors could incorporate more information regarding specific parenting practices as a protective factor.

We revamped this content, added more empirical studied/literature to focus on parenting as a protective factor for this population.

The paragraph on intersectionality and the bioecological model could come prior to the discussion of risk factors, especially given that it would help introduce the notion that there are risk and protective factors at multiple levels of youths’ social ecologies.

We moved the theoretical framework before the risk and protective factors to introduce the idea that both risk and protective factors exist in a unique context and across social domains.

Method

It would be helpful if the authors’ defined the acronym ACASI and the methodology or refer the reader to prior publications using NSDUH data/methods so that the methods of data collection are clear.

Added a footnote to define ACASI:

ACASI (audi computer-assisted self-administered interview) equipment technique allows standardization of the way in which questions are asked and who is asking them, and it eliminates interviewer interpretation of responses. The perceived anonymity of this type of interview may make respondents feel more at ease in reporting behaviors that are socially undesirable and less likely to embellish responses for socially desirable behaviors

With this citation:

Kissinger, P., Rice, J., Farley, T., Trim, S., Jewitt, K., Margavio, V., & Martin, D. H. (1999). Application of computer-assisted interviews to sexual behavior research. American Journal of Epidemiology, 149(10), 950–954.

It would also be useful for the authors’ to clarify whether the NSDUH measures (e.g., religiosity, positive parenting, school engagement, and activities) were based on pre-existing measures of these constructs or created questions as well as which specific items from the NSDUH were used for the purposes of these analyses. This will help other researchers with replicability.

Our measures of religiosity, positive parenting, school engagement, and activities were created based on items from the NSDUH, for the purposes of this analysis. However, we used items to create scales that have been used in prior studies with the NSDUH. For example, other researchers have used the NSDUH to assess religiosity (Ford & Hill, 2012) and positive parenting (Goodwill, 2020) in the same way as this study. In our revised manuscript, we are more specific about what specific items were used for each subscale.

Results

The authors’ should add in information regarding missingness and what was done with missing data (e.g., listwise deletion, multiple imputation).

We now specify the amount of missing data and how it was handled in our analysis:

“Missing data ranged from 0 to 19.1% depending on the study variable. Unadjusted and adjusted logistic regressions were modeled using the sample with complete data (i.e., listwise deletion).”

Further, we adjusted Tables 2 and 3 to accurately reflect the n for each separate model.

More rationale is needed regarding to the reasoning for running adjusted and unadjusted logistic regression models. It would also be helpful for the authors’ to add a sentence or two to orient the reader to the interpretation of odds ratios.

We now specify: “First, univariate analysis was

conducted to investigate the associations between each of our hypothesized risk and protective factors, separately, with suicidal thoughts and behaviors. Following this, we tested three adjusted logistic regression models (i.e., in a multivariate analysis) that were run separately for each outcome (suicidal ideation, suicide planning, and suicide attempts). The adjusted logistic regression allowed us to determine the influence of each risk and protective factor while adjusting (i.e., controlling) for the other predictor variables in our model. 

There appears to be a paragraph included in error beginning online 286 with “This section may be divided by subheadings.”

We removed this content as it was included by mistake:

This section may be divided by subheadings. It should provide a concise and precise description of the experimental results, their interpretation, as well as the experimental conclusions that can be drawn.

Discussion

The authors should be cautious in overfocusing their discussion on factors that did not emerge as significant in their final multivariate logistic regression model. Variables that were statistically significant in the single logistic regression models could have emerged as significant due to spurious associations.

We revised the discussion to focus on the significant factors in the analyses.

The authors’ description of their multivariate model should come at the beginning of the discussion and not towards the end.

We moved the description of the multivariate model to the first paragraph of the discussion to set the tone.

Given that this present sample involves justice-involved youth, some attention to of the role that involvement plays in the present results would strengthen the Discussion section.

We added more empirical studied/literature about the role of system involvement and mental health problems for this population.

Given that a single-item measure was used for youth depression severity, this should be noted as a limitation.

We added the single-item depression measure as a limitation.

It would also be great for the authors’ to have a separate future directions paragraph. For example, the authors’ suggest training on positive parenting skills as a future direction. How could this be fit into research on treatment modalities specific to justice-involved youth (e.g., multisystemic therapy)?

We created a separate paragraph for future research and noted the need to focus on treatment modalities specific to this population of youth. of We also described studies to develop positive parenting training.

Reviewer 2 Report

It is an interesting manuscript, but it is necessary to improve some aspects. 
Introduction
Specify in the corresponding statements that they are statements based on data collected in the USA.
Find an academic basis for the statement in lines 42 and 43.
At line 45 there appears to be a parenthetical citation, adjusting to the style of the journal.
Specify the meaning of ACEs.
Why was reference 42 not included in line 124? Adjust the citation numbering as appropriate, as well as the references.
The objective expressed in the abstract and in lines 65-67 seems confused using the word “constructs”. Why not establish directly these are risk and protective factors?
Regarding the outcome of suicidal behavior, it is necessary to clarify that it is suicidal thoughts, plans or attempts at some time in life.

Materials and method
Regarding the participants, the sentences in lines 171 and 172 are confusing. Improve the writing. It is better to mention those who participate in only one assistance program and those who participate in two or more assistance programs.
In line 172, an n=845 is mentioned, when in the abstract and in line 167 an N=513 was mentioned. Clarify this.
It is necessary to clarify the procedure to establish the severity of depression.
In line 213 there is a missing data. ( =.77). In line 217 and 224 there are missing data, too.

Results
In Table 1 it seems that values of lost surveys are shown. This needs to be clarified in the methodology.
Table 2 is not complete. It is necessary to include the complete table.
Lines 286-288 reflect a comment, not a result.

Discussion
There is a typo in line 302.
The comparison made in lines 304 and 305 is imprecise. Regarding reference 56, a higher prevalence of suicidal ideation and plans is to be expected in the present study, since the Mid-Atlantic study asked about suicidal plans in the last 12 months, when in the present study it asked about suicidal behavior at some time in life. Reference 55 does not clarify how the evaluation of suicidal behaviors in the Midwestern population was made, in addition to including in the same data ideation and suicide attempt, not planning as stated in the manuscript. Therefore, the conclusions in lines 308-311 are not supported. The reported prevalence of suicidal ideation and suicide attempt in this study are lower than those reported in a systematic review by Teplin et al., 2015. It is necessary to deepen the discussion on the prevalence obtained in this study.
The statement in lines 321-323 is inaccurate, as reference 61 reported that positive parental relationships decreased suicidal ideation and attempt.
The claims in lines 323-330 lack empirical support. It is necessary to include it. I recommend further establishing the importance of family relationships in the face of suicidal behavior. This can be found in other minority ethnic groups.
